# Comparison of the Genomic Profile of Cancer Stem Cells and Their Non-Stem Counterpart: The Case of Ovarian Cancer

**DOI:** 10.3390/jcm9020368

**Published:** 2020-01-29

**Authors:** Elena Laura Mazzoldi, Anna Pastò, Giorgia Pilotto, Sonia Minuzzo, Ilaria Piga, Pietro Palumbo, Massimo Carella, Simona Frezzini, Maria Ornella Nicoletto, Alberto Amadori, Stefano Indraccolo

**Affiliations:** 1Immunology and Molecular Oncology Unit, Veneto Institute of Oncology IOV - IRCCS, 35128 Padova, Italy; elena.mazzoldi@libero.it (E.L.M.); annapasto.phd@gmail.com (A.P.); giorgia1210@hotmail.it (G.P.); ilaria.piga1992@gmail.com (I.P.);; 2Department of Surgery, Oncology and Gastroenterology, University of Padova, 35128 Padova, Italy; soniaanna.minuzzo@unipd.it; 3Medical Genetics Unit, Fondazione IRCCS Casa Sollievo della Sofferenza, 71013 San Giovanni Rotondo, Italy; p.palumbo@operapadrepio.it (P.P.); m.carella@operapadrepio.it (M.C.); 4Medical Oncology 2, Veneto Institute of Oncology IOV - IRCCS, 35128 Padova, Italy; simona.frezzini@iov.veneto.it (S.F.); ornella.nicoletto@iov.veneto.it (M.O.N.)

**Keywords:** ovarian cancer, cancer stem cell, genetic heterogeneity, SNP array

## Abstract

The classical cancer stem cell (CSC) model places CSCs at the apex of a hierarchical scale, suggesting different genetic alterations in non-CSCs compared to CSCs, since an ill-defined number of cell generations and time intervals separate CSCs from the more differentiated cancer cells that form the bulk of the tumor. Another model, however, poses that CSCs should be considered a functional state of tumor cells, hence sharing the same genetic alterations. Here, we review the existing literature on the genetic landscape of CSCs in various tumor types and as a case study investigate the genomic complexity of DNA obtained from matched CSCs and non-CSCs from five ovarian cancer patients, using a genome-wide single-nucleotide polymorphism (SNP) microarray.

## 1. Cancer Stem Cell (CSC) Theory

It has been established and it is well accepted that tumors are composed of heterogeneous cell populations. According to the classical hierarchical model, tumors arise from a small population of cells, called CSCs, derived from the malignant transformation of a normal stem cell. One of the main features of stem cells is their unlimited proliferative potential to sustain renewal and repair needs of normal tissues after injury. However, following exposure to environmental carcinogens or due to stochastic effects, normal stem cells can accumulate genetic mutations in cancer-associated genes (e.g., oncogenes and onco-suppressor genes), as well as defects in the DNA repair machinery. These mutated stem cells can become resistant to apoptosis and undergo malignant transformation in CSCs [1]. According to this model, CSCs inherit all the key features from their normal stem counterparts, including the unlimited proliferation rate and self-renewal capability. This implies that CSCs can accumulate a number of genetic or epigenetic changes, subsequently inherited by tumor cells derived from CSC asymmetric division and differentiation (Figure 1A, left panel) [2,3]. Due to the genetic heterogeneity of cancer, it is likely that the mutational burden of CSCs varies among different tumors, as occurs in non-CSCs. On the other hand, other studies indicate that CSCs develop mechanisms to lower radical oxygen species (ROS) accumulation and to extrude drugs, thus reducing the risk of damage to their genetic content [4]. These features let us speculate a reduced number of genetic alterations in CSCs compared to non-CSCs (Figure 1A, right panel). Moreover, non-CSCs could possibly accumulate private mutations—defined as mutations not shared by CSCs—during subsequent rounds of proliferation of trans-amplifying cells and bulk tumor cells (Figure 1A). 

In contrast to the hierarchical model, an alternative theory poses CSC as a functional state of a tumor cell (reviewed in [5]). According to this alternative model (Figure 1B, left panel), stemness can not only be a cellular intrinsic property but also the result of extrinsic stimuli and of the cross-talk between CSCs and the complex network of cells, matrices, and vesicles that cooperate to maintain a tumor permissive microenvironment [6]. For example, in a study on breast cancer in which cell lines were used as models, only one subpopulation of cells out of three (stem cell-like, basal-like, and luminal-like cells) was tumorigenic *in vivo*, whereas in another setting, in which irradiated stromal cells were provided, all three considered subpopulations where equally tumorigenic [7]. Within the tumor microenvironment, CSCs, as their normal adult counterpart, reside in specialized niches, since they need to receive cues from the surrounding cells in order to activate and maintain their stemness program (self-renewal, proliferation and apoptosis resistance) and stemness regulating pathways (i.e., Notch, Wnt/β-catenin and Hedgehog). The niche consists of different cell types, including tumor-associated fibroblasts (TAFs), endothelial cells, pericytes, and immune cells, especially tumor-associated macrophages (TAMs) and myeloid-derived suppressor cells (MDSCs), as well as non-cellular elements, such as the extracellular matrix and the complex network of growth factors and cytokines [6,8]. All these components regulate stemness in different ways, i.e., by secreting cytokines which activate signaling pathways and stimulate self-renewal [6], or by promoting epithelial-to-mesenchymal transition [9]. 

According to this alternative model, the genetic profile of CSCs and non-CSCs is predicted to be quite similar (Figure 1B, right panel). 

## 2. Genetic Profile of Cancer Stem Cells Versus the Bulk of Tumor Cells

In the last decade, CSCs have been identified in most tumor types (both solid and liquid) based on the expression of specific surface markers, lack of certain differentiation markers and high tumorigenic potential. In addition, CSCs have the ability to grow in vitro under serum-free conditions as rounded structures called spheroids and undergo asymmetric cell division, thus originating one daughter cell (maintaining stemness features) and one differentiated cell.

To date, only a few attempts have been made to describe the cytogenetic complexity of CSCs, mainly comparing their karyotype to that of non-CSCs (Table 1). Most studies analyzed either tumor cell lines or a small number of primary samples, due to the difficulty of obtaining enough genomic DNA from CSCs. Moreover, CSC isolation was often not performed according to stemness marker expression, but rather by using in vitro specific cell culture protocols. Along this line, Gasparini and colleagues performed a complete cytogenetic characterization of sphere-growing stem-like cells from six different cell lines of melanoma, breast, lung, and ovarian cancer [10]. 

Results indicated a more rearranged karyotype of CSCs, compared to the parental cell lines maintained under standard adhesion culture conditions. CSCs showed a higher number and complexity of chromosomal alterations in all cell lines evaluated. One bias of this study, acknowledged by the authors, was the use of immortalized cell lines and in vitro culture techniques for CSC isolation and expansion. Indeed, CSC-enriched spheroids are mainly composed of highly dividing progenitors that could affect results of genetic analysis. In any case, the main finding of this study was that growth culture conditions affect the genetic landscape of tumor cells.

In another study, Lee and colleagues demonstrated that glioblastoma (GBM) cells from primary tumor samples, cultured in serum-free spheroid-forming conditions, harbored extensive genotype similarity to parental tumor cells [13]. In contrast, when cultured in serum-supplemented media, glioblastoma cells underwent genomic rearrangements in terms of loss of heterozygosity (LOH), pseudo-tetraploidy and chromosomal deletions, suggesting a key role of in vitro culture conditions in the acquisition of genetic instability. Unfortunately, it is unknown whether these in vitro findings can be relevant to CSCs from patients.

Few other studies were performed in different tumor types, including GBM, breast cancer, head-and-neck squamous cell carcinoma (HNSCC), and bladder cancer, in order to compare either FACS-sorted CSCs or spheroid-forming cells with either their non-CSC counterpart or the tumor bulk (Table 1). Piccirillo et al. analyzed 12 primary GBM tumor sample-derived neurospheres by single nucleotide polymorphism (SNP) array and found that four out of 12 presented at least three copy number alterations (CNAs). Then, the authors analyzed one of these samples in detail, by using whole-exome sequencing (WES) and single-cell analysis. They concluded that the selected somatic mutations and CNAs highlighted in the primary tumor were also present in the derived neurospheres. Moreover, some genetic alterations were subclonal and correlated with different tumorigenic potential of single tumor cells in mice. Thus, conclusions suggest that CSCs and bulk tumor cells share the same genetic alterations but GBM presents substantial intra-tumor genetic heterogeneity [11].

In an independent study, Pesenti et al. obtained spheroids from three out of 10 primary GBM tumor samples and analyzed them by array comparative genomic hybridization (aCGH). In each case, they observed that tumor cells and their derived spheres shared the same genetic alterations [12], in line with previous findings [19]. Altogether, all these studies demonstrated that neurospheres share mutations and alterations with parental GBM cells.

Three other studies focused on breast cancer. Kleverbring and colleagues compared bulk tumor cells and spheroids in ten primary samples, sorted CD44^+^/CD24^−^ and CD44^−^/CD24^−^ cells, and sorted aldehyde dehydrogenase positive (ALDH^+^) and negative (ALDH^−^) cells in two additional samples, by WES, and validated their findings by ultra-deep amplicon sequencing [14]. In all the analyses, the Authors observed that CSCs and non-CSCs shared most mutations; thus, they concluded that the existence of CSCs and non-CSCs is the result of a continuous dynamic transition between a stem- and a non-stem functional state due to cell plasticity, rather than being distinct cell populations irreversibly characterized by different genomic landscapes.

Tiran et al. compared FACS-sorted CD44^+^/CD24^−^ and ALDH^+^ cells with bulk tumor cells, obtained from pleural effusions, by low-coverage whole genome sequencing (WGS), thus demonstrating that FACS-sorted CSCs and the bulk tumor shared the same alterations [15].

Tong and colleagues compared, by WGS, MDA-MB-231 cells maintained either in adherent culture conditions or in spheroid-forming conditions and validated their findings by target deep DNA sequencing [16]. They demonstrated that the observed SNVs were characterized by a similar allelic frequency in both cell culture conditions, thus indicating that no variant was specifically associated with the stem status in their experimental model. Rather, spheroid-forming cells distinguished themselves from the adherent counterpart for a distinct gene expression profile.

Once again, putative breast CSCs, either sorted on the basis of surface markers or enzymatic activity or enriched by serum-free conditions, seem to be genetically similar to the bulk tumor cells, even though they differ for the functional properties or their transcriptomic profile.

Finally, sporadic studies reported the CSC genomic profile of other tumor types. Salazar-Garcia et al. tried to rebuild the CSC evolutionary history by WES analysis of ALDH^+^ and ALDH^−^ cells FACS-sorted from four HNSCC primary tumors, as well as normal cells. By evaluating LOH, they hypothesized that in some patients CSCs derived from the neoplastic transformation of normal tissue, whereas in other patients they derived from dedifferentiation of tumor cells. Indeed, if a variant was present in normal cells and in CSCs in heterozygosity, but heterozygosity was lost in the differentiated tumor cells, it is likely, according to the authors, that tumor cells derived from CSCs, which derived from normal tissue. On the contrary, if a variant was present in heterozygosity in both normal and tumor cells, but not in CSCs, such cells probably derived from tumor cell dedifferentiation. Thus, in HNSCC, the authors observed a certain extent of genetic difference between CSCs and non-CSCs.

In one case of invasive urothelial bladder carcinoma, Prado et al. FACS-isolated CD44^+^/CD49f^+^/EpCAM^+^ CSCs from both primary tumor and lymph node metastases and analyzed them by WES, comparing results with the bulk tumor cells and normal lymph nodes [18]. They found 51 SNVs, of which the majority was shared by CSCs and tumor cells. Only a small number were uniquely found in the bulk tumor cells or in CSCs. The authors concluded that the SNVs unique in bulk were the result of clonal evolution and the SNVs unique in CSCs belonged to a quiescent subpopulation that had not yet generated a progeny big enough to be detectable among the other bulk cells.

## 3. The Case of Ovarian Cancer

Ovarian cancer is the fifth most frequent female cancer and the most common cause of death from gynecological tumors [20]. Epithelial ovarian cancer (EOC) comprises almost 90% of all cases [21]. Even if several models have been proposed to explain EOC pathogenesis [22], origin of EOC is still debated. Indeed, although serous tubal intraepithelial carcinoma (STIC)—a non-invasive tumor formed preferentially in the distal fallopian tube epithelium—has been considered by the scientific community as a precursor of high-grade serous carcinoma, in many cases it is unclear whether advanced stage disease results from progression from an early stage.

Known risk factors for ovarian cancer include the number of ovulations [23], inflammatory conditions [23,24], factors of hormonal nature [25], and genetic predisposition. Indeed, women with Lynch syndrome and with germline mutations in BRCA-1 and BRCA-2 have increased lifetime risk of several cancers, including EOC [19].

As for other tumors, CSCs in ovarian cancer has been identified according to the expression of specific surface markers. One of the most widely used markers is CD133 or Prominin, introduced by Curley and colleagues who demonstrated the higher tumorigenic potential of CD133^+^ cells, compared to CD133^−^, isolated from primary samples of EOCs and injected into immunocompromised mice [26]. However, more recently, the use of CD133 has been debated and novel markers were added for CSCs identification, such as ALDH, a detoxifying enzyme that enables CSCs to survive chemotherapeutic drugs. Recently, CD44 and CD117 (c-kit) have been proposed by Zhang and colleagues as markers for CSCs in EOCs [27]. These results were also confirmed by Pastò et al., who demonstrated that CD44^+^CD117^+^ double positive cells presented all canonical features of CSCs including ability to grow as spheroids, expression of stemness-associated markers (e.g., Nanog, Sox2 and Oct4), expression of multidrug-resistant pumps involved in drug extrusion, and high tumorigenic potential when injected into immunocompromised mice [28].

With regard to ovarian cancer, it is unknown whether CSCs disclose a genetic fingerprint similar to non-CSCs. We consider this question important, as its answer can help to discriminate whether epithelial ovarian CSCs fit the standard CSC hierarchical model or whether they are to be considered a functional state of tumor cells, as has been advanced for other tumor types. This lack of knowledge motivated our choice to investigate in a pilot experiment the genomic complexity of DNA obtained from matched CSCs and non-CSCs from ovarian cancer patients using a genome-wide SNP microarray. We performed karyotype profile analysis on CSCs isolated from human primary cultures of EOC established from high grade serous ovarian cancer ascitic fluid samples. The study was approved by the local ethical committee. CSCs were FACS-sorted as CD44^+^CD117^+^ from the ascitic effusions of eight patients and the extracted DNA was analyzed by high-density SNP arrays (CytoScan^®^ HD Array, Affymetrix, Santa Clara, California, USA) and compared to CD44^+^CD117^−^ cells (non-CSCs). The essential clinical features of these patients are summarized in Table 2.

In three out of eight pairs of CSCs and non-CSCs analyzed quality of the DNA samples was too low and precluded comparison of the genetic fingerprints of the two subpopulations. In the remaining five pairs (49, 84, 98, 101, and 106), SNP analysis was successful and did not disclose any genetic difference between CSC and non-CSC DNA samples, with the exception of a mosaic rearrangement on chromosome 2 (arr[GRCh37] 2p21p11.2(45744391_84671244)× 2-3) detected in sample 84_CSC which was not detected in the matched 84_non-CSC sample. Additional genetic studies, such as mutation profiling of matched CSC and non-CSC samples could not be performed, due to limited amount of tumor gDNA. Finally, in the case of patient #49, it was possible to obtain two samples of ascitic fluid at a 12 month-interval (#49_III and #49_V) and SNP array analysis was performed in both cases confirming an identical genotype of the CSC and non-CSC subpopulations. Notably, although within each pair no or only marginal genetic differences were found, each tumor sample presented multiple genetic alterations in terms of LOH, deletions or amplifications, compared with normal genomes. The representative profile of one of these samples is shown in Figure 2.

In summary, although the number of pairs analyzed was low, we found that patient-derived ovarian CSCs present very similar genetic features compared with non-CSCs. Therefore, it is likely that epigenetic differences account for the marked functional differences between these two sub-populations described elsewhere [28].

## 4. Conclusions

The classical CSC model places CSCs at the apex of a hierarchical scale, implying different genetic alterations in non-CSCs compared to CSCs, since a number of cell generations and time intervals separate CSCs from the more differentiated cancer cells that form the bulk of the tumor (Figure 1). In addition, CSCs seem to be endowed with more efficient DNA repair mechanisms, which partially shield their genome from genotoxic events. In this study, we tested this prediction in the case of ovarian cancer. Our results suggest that CSCs are genetically very similar to more differentiated cancer cells. Altogether, our findings agree with the majority of previous studies in other tumor types (Table 1) and support the alternative theory which poses CSC as a functional state of a tumor cell rather than a specific cell type.

How is this functional state of CSCs induced? Although a full answer to this question is currently not possible and is beyond the scope of this article, several intriguing hypotheses deserve to be mentioned. In a recent work, Canova described three stemness-associated genes: Nanog, Sox2, and Oct4 [29]. Each one controls the differentiation into a specific cell lineage, repressing the alternative; when co-expressed, they actually block differentiation in all lineages, thus promoting a stemness phenotype. We and others have found that ovarian CSCs co-expressed Nanog, Sox2, and Oct4, whereas expression of these genes is reduced or absent in tumor non-CSCs. Thus, stemness seems to be maintained because the differentiation pathways are blocked, rather than because the stemness pathways are activated. In addition, it has been demonstrated that some cells (i.e., hepatocytes and pancreatic islet cells) can de-differentiate under specific stimuli [30]. These results imply that tumor cells (non-CSCs) could potentially be forced to stemness by pressure from surrounding cells (i.e., within the tumor microenvironment) or stress conditions (i.e., anti-tumor drug treatment, metabolic substrate, or oxygen restriction). In order to survive, these cells block differentiation pathways and de-differentiate into CSCs, thus acquiring the ability to enter quiescence, overexpress multi-drug resistant pumps to extrude toxic compounds, or activate altered metabolic pathways, all canonical features of CSCs [28,31].

In conclusion, our results, albeit limited to a small number of cases, support the alternative CSC model shown in Figure 1 and suggest stemness as a dynamic state, a state of plasticity between tumor cells and CSCs. The genetic similarity of CSCs and non-CSCs should be taken into account in the development of successful new therapeutic approaches for ovarian cancer.

## Figures and Tables

**Figure 1 jcm-09-00368-f001:**
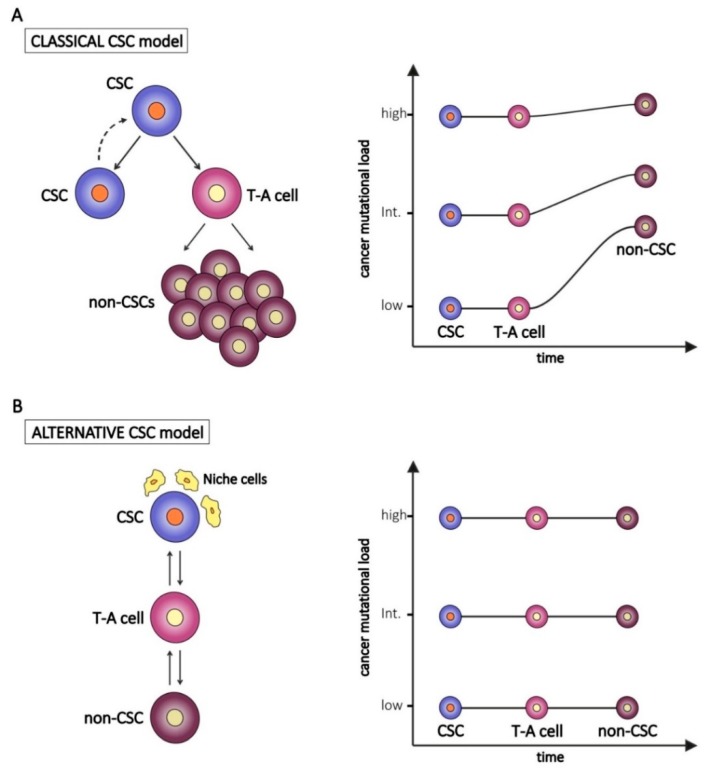
The classical (**A**) and the alternative (**B**) “Cancer Stem Cell (CSC)” models and hypothetical implications for the tumor mutational burden of transient-amplifying (T-A) and bulk tumor cells (non-CSCs). In the right panels, examples of CSC with different levels of genetic alterations (low, intermediate, high) are represented.

**Figure 2 jcm-09-00368-f002:**
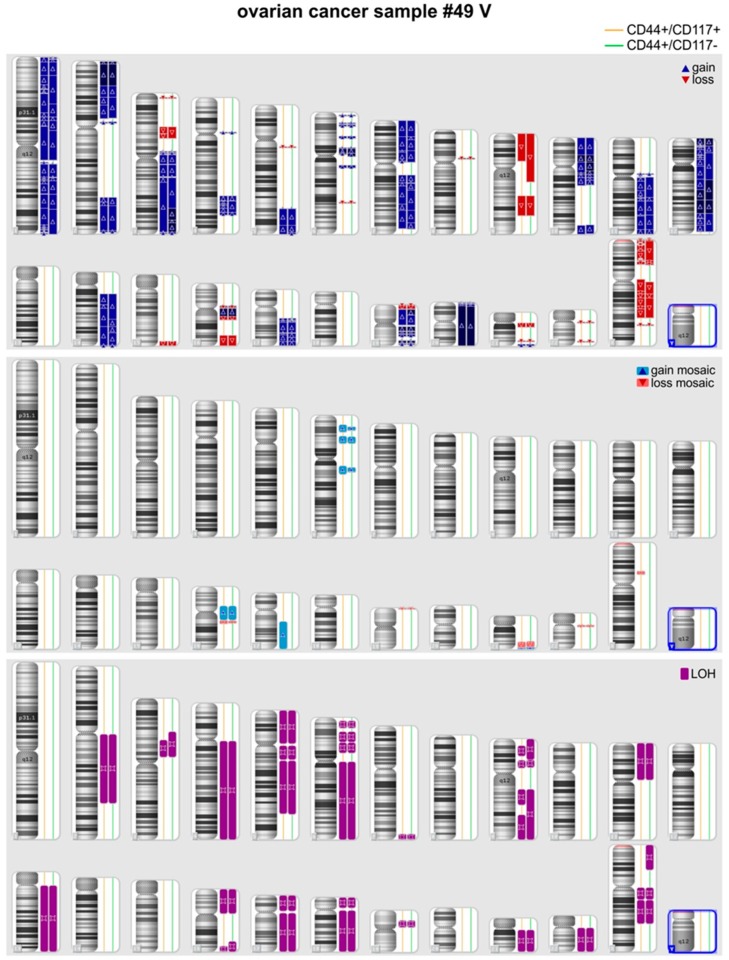
Comparison of chromosomal alterations in CSCs versus non-CSCs by Cytoscan SNP-array in one representative ovarian cancer patient (#49 V).

**Table 1 jcm-09-00368-t001:** Studies on genetic alterations in CSCs.

Cancer Type	Materials	Methods	Main Conclusions	Reference
GBM	Primary tumor cells	SNP array, WES, single-cell analysis	CNAs found in primary tumor cells are also present in neurospheres in different subclones	Piccirillo S.M. et al., 2015 [11]
GBM	Primary tumor cells	aCGH	High similarity between GBM mass and spheroids	Pesenti C. et al., 2019 [12]
GBM	Primary tumor cells	SKY; SNP array	Cells cultured in serum-containing medium underwent genomic rearrangements, while spheroids did not	Lee J. et al., 2006 [13]
Breast	Primary tumor cells	WES; ultra-deep amplicon sequencing	Mutations are shared between tumor bulk and spheres	Klevebring D. et al., 2014 [14]
Breast	Tumor cells from pleural effusions	low-coverage WGS	Same alteration in sorted CSCs and bulk tumor	Tiran V. et al., 2017 [15]
Breast	MDA-MB-231 cell line	WGS; target deep sequencing	No differences in VAF between monolayer and spheres	Tong M. et al., 2018 [16]
HNSCC	Primary tumor cells	WES	From LOH analysis, it is hypothesized that CSCs may originate either from normal tissue or from tumor cell dedifferentiation	Salazar-Garcia L. et al., 2018 [17]
Bladder	One primary tumor and lymph node metastases	WES	SNPs are mainly shared by sorted CSCs and bulk tumor cells; a small number is enriched either in CSCs or in bulk cells	Prado K. et al., 2017 [18]
Various	Cell lines	SKY	More rearranged genotype of spheres compared to parental cell lines	Gasparini P. et al., 2010 [10]

Abbreviations: CSCs, cancer stem cell; GBM = glioblastoma multiforme; SNP = single nucleotide polymorphism; WES = whole exome sequencing; can = copy number alteration; aCGH = array comparative genomic hybridization; SKY = spectral karyotyping imaging; WGS = whole genome sequencing; VAF = variant allele frequency; HNSCC = head and neck squamous cell carcinoma; LOH = loss of heterozygosity.

**Table 2 jcm-09-00368-t002:** Clinical features of EOC patients involved in the study.

Sample	Histotype	Stage	Grade	Chemotherapy	CD117 enrichment *
49 III	Serous-papillary	3C	3	Yes	7.08
49 V	Serous-papillary	3C	3	Yes	2.41
84 IV	Serous-papillary	3C	3	Yes	4.02
98	Serous	3B	1	No	21.53
101	Serous	3C	3	Yes	4.37
106	Bilateral serous-papillary	4	3	No	5.99

* CD117 mRNA expression (fold change) between CD44^+^CD117^+^ and CD44^+^CD117^−^ FACS-sorted populations. This parameter is used to check by an orthogonal technique CSC enrichment after sorting, EOC, Epithelial ovarian cancer.

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
