# Peer review of "Comparison of the Genomic Profile of Cancer Stem Cells and Their Non-Stem Counterpart: The Case of Ovarian Cancer"

_jcm, 2020, doi:10.3390/jcm9020368_

Round 1

Reviewer 1 Report

The revisions by the authors have greatly improved the manuscript.

The new section 3 is missing indentions for the paragraphs.

Author Response

We thank the reviewer for appreciating our revised work and added indentions to section 3.

Reviewer 2 Report

The manuscript by Elena Laura Mazzoldi and colleagues reviews existing literature on the genetic landscape of CSCs and their non-CSCs counterparts in various tumor types. The review is well written, it is concise and summarizes findings of a limited number of studies available on this topic.  In the second part, it focuses on ovarian cancer and includes data of a pilot experiment performed by the investigators.

I have these comments:

The authors may consider changing the article title to clearly indicate the article is focuses on comparisons between CSCs and non-CSCs (and TA cells in some cases). The current title invokes rather broader associations. The investigators’ findings are not supported by any data. I understand, that the results are mostly negative (no genetic difference between CSC and non-CSC DNA samples, with the exception of a mosaic rearrangement on chromosome 2 detected in one sample CSC which was not detected in the matched non-CSC sample), but still, the data should be presented in at least one figure. The authors state (lines 205-207): Even if several models have been proposed to explain EOC pathogenesis [21], etiology of EOC is still substantially unknown. Indeed, no precancerous lesion has been identified and it is unclear whether advanced stage disease is a result of progression from an early stage.” However, serous tubal intraepithelial carcinoma (STIC), a noninvasive tumor lesion formed preferentially in the distal fallopian tube epithelium, has been heavily proposed in the research community as a precursor for high-grade serous carcinoma, the most frequent and deadliest type of EOC. Could authors comment on this, please?

Author Response

We thank the reviewer for the constructive criticisms and improved the manuscript accordingly. Specifically:

The title was changed and is now more specific. We added some more experimental details in text as well as a table and a figure to support our statements. We thank the reviewer for pointing out this inaccuracy and corrected the sentence regarding precursors of high-grade serous carcinoma according to his/her suggestions.

Round 2

Reviewer 2 Report

The authors answered all my comments adequately and promptly.

This manuscript is a resubmission of an earlier submission. The following is a list of the peer review reports and author responses from that submission.

Round 1

Reviewer 1 Report

In the study entitled "Genomic characterization of ovarian cancer stem cells", authors reviewed existing literature on the genetic landscape of Cancer Stem Cells (CSCs) in various tumor types and investigate the genomic complexity of DNA obtained from matched CSCs and non-CSCs from five ovarian cancer patients by using a genome-wide SNP microarray.

  The manuscript is not presented in an acceptable format. It includes some draft sentences. Such an example from the materials and methods section is given below:

"Materials and Methods should be described with sufficient details to allow others to replicate and build on published results. Please note that publication of your manuscript implicates that you must make all materials, data, computer code, and protocols associated with the publication available to readers. Please disclose at the submission stage any restrictions on the availability of materials or information. New methods and protocols should be described in detail while well-established methods can be briefly described and appropriately cited.

Research manuscripts reporting large datasets that are deposited in a publicly available databases should specify where the data have been deposited and provide the relevant accession numbers. If the accession numbers have not yet been obtained at the time of submission, please state that they will be provided during review. They must be provided prior to publication. Interventionary studies involving animals or humans, and other studies require ethical approval must list the authority that provided approval and the corresponding ethical approval code.

It does not add any significant information to the literature. It is very preliminary to draw conclusions presented in this manuscript with the data from only 5 cases. The manuscript might only be considered as a case study if presented in such format. It is not suitable to be published in Journal of Clinical Medicine in its present form. It is also suggested to have a native speaker to check the manuscript for grammatical and spelling inconsistencies.

Author Response

We thank the reviewer for his/her comments and corrected the format of the manuscript in particular the Materials and Methods section, as suggested. We are well aware of the main limitation of our pilot study, i.e. the low number of cases analyzed. Indeed, we remarked this both in the Introduction (line 248), Results (lines 340-1) and in the Conclusions (line 382). Moreover, it should be considered that the co-primary aim of our study was to review previous knowledge on this topic, as is clear from the long and structured Introduction. Finally, the text has been edited by Ms Christina Drace - a professional editor -for the English language.

Reviewer 2 Report

There are a number of one sentence paragraphs that should be combined into longer paragraphs. The methods should be “Fluorescence activated cell sorting” not “flow cytometry” since the cells are collected afterwards. The manuscript references a mosaicism in sample #84_CSC but the data does not appear in table 2. There does appear to be a gain of function mosaicism in sample #49 in figure 2.

Author Response

We thank the reviewer for the comments, modified in the Methods the sentence indicated and had the entire manuscript reviewed by a mother English-speaking editor. We also corrected the text (lines 327-330) in order to briefly describe the mosaicism in sample #84 (which is not in table 2).

Reviewer 3 Report

In the present study, the authors have examined signature gene TP53 and its relevance with stemness and tumor plasticity in ovarian cancer. Results from a set of 12 primary tumors collected from 12 different tissues were studied and stratified into two classes characterized by recurrent mutations (M class) or recurrent copy number alteration (C class). Role of TP53 mutations and multiple recurrent chromosomal gains and losses, has been studied as C class tumor and extended for the genomic complexity of DNA obtained from matched CSCs and non-CSCs from ovarian cancer patients. I have following comments for the authors, which should be addressed for a comparative assessment.

Cancer stemness and TP53 mutations are quite complex for inducing the tumor biological implications. The authors should demonstrate these issues by using one additional technique to corroborate these results. How these mutations differ at functional levels? Do they look on other biological pathway or induce signal transduction in CSC versus non -CSC from ovarian cancer? The authors should include the limitations of SNP-microarray and address their data by other multiplex technology for a comparative assessment. How these results carry biological significance in vitro and in vivo animal models? They should analyze their results by running a comparative data analysis with other published work by other investigators. The sentences 149, 193 and few other in the text need to be corrected as they look to be prepared for a scientific presentation rather than a manuscript for publication. In addition, lines 314-325 do not make sense in the text. It looks that has been copied from Instruction for Authors or Users of the journal or to operate an instrument. These should be fixed. Do such stemness, TP53 and a state of plasticity data have similarity with other tumor type in samples from human cancer subjects? Introduction is too long and at time loses the grip over the primary objectives of the manuscript. This should be condensed.

Author Response

We thank the reviewer for the comments, but would like to point out that we did not examine the signature of gene TP53 and its relevance with stemness, as the reviewer states. This is rather what Ciriello G. et al. (i.e. the Authors of reference 31) did in their study, which we quoted in our work. There must have been a misunderstanding, and we tried to better formulate the sentence which might have generated this issue (lines 242-248).  With regard to the other experiments proposed by the reviewer, due to the limited amount of genomic DNA from purified CSCs it was not possible to cross validate results of the SNP-microarray analysis. Finally, sentences at lines 149 and 193 have been modified and lines 314-325 which did not make sense have been deleted. Introduction is indeed long but this was a primary objective of our manuscript, as we wished to review the scientific literature on this topic before presenting our own results.

Round 2

Reviewer 1 Report

The paper in it's present format does not merit publication in the Journal of Clinical Medicine. Authors still did not answer questions raised by the previous reviewers significantly. It does not add any significant value to the literature.

Reviewer 3 Report

The authors have examined signature gene TP53 and its relevance with stemness and tumor plasticity in ovarian cancer. Results from a set of 12 primary tumors collected from 12 different tissues were studied and stratified into two classes characterized by recurrent mutations (M class) or recurrent copy number alteration (C class). The revised version still lacks clarity and scientific information. The authors should address following issues carefully for improving the quality of the manuscript.

The authors have stated and expressed their inability to confirm their initial results on TP53 and associated tumor biological events because of limited quantity of DNA from CSCs and non-CSCs. The authors should collect a few more samples, generate cDNA, perform linear amplification to get cDNA in sufficient quantity for a panel of assays/experiments. It is not safe to rely on previously published work on a different cohort (?) of samples of Ref 31 to conclude the present set of results. Are the present sets of data done independently and concur the results shown in reference 31? If yes, it should be clearly described in the manuscript. If not, the authors should perform a new set of experiments and establish any molecular or cellular link with TP53 mutations and discuss critically with the reference 31. In addition, the present set of data of TP53 mutations and multiple recurrent chromosomal gains and losses, should be confirmed by other molecular/cellular biological assays/techniques as these are quite complex issues and may lead to interpret just opposite to what it may be happening biologically in situ for the genomic complexity of DNA obtained from matched CSCs and non-CSCs from ovarian cancer patients. Cancer stemness and TP53 mutations are quite complex processes in the field of oncogenesis. The authors should work on these issues by using some techniques such as RT-qPCR or in situ hybridization if these helps establishing the role of TP53 and cancer stemness. As mentioned above, if these data concord with this reference 31, please establish the degree of concordance and add this information in Results with adequate statistical analysis. The authors should demonstrate the genetic link between TP53 mutations and cancer cell plasticity/stemness. How these mutations differ at functional levels in terms of signaling ? Do they look on other biological pathway or induce signal transduction in CSCs versus non -CSCs from ovarian cancer? These are important issues, which need to be undertaken and studied carefully. The authors should include the limitations of SNP-microarray and address their data by other multiplex technology for a comparative assessment. How these results carry biological significance in vitro and in vivo animal models? These are very important and critical check points, which need to be addressed in at least few representative samples.       They should analyze their results by running a comparative data analysis with other published work by other investigators. Introduction is still too long and at time loses the grip over the primary focus of the manuscript. This should be condensed.